# Loco-Regional Anaesthesia during Standing Laparoscopic Ovariectomy in Equids: A Systematic Review (2003–2023) of the Literature

**DOI:** 10.3390/ani14162306

**Published:** 2024-08-08

**Authors:** Giada Giambrone, Giuseppe Catone, Gabriele Marino, Enrico Gugliandolo, Renato Miloro, Cecilia Vullo

**Affiliations:** 1Department of Veterinary Sciences, University of Messina, Polo SS. Annunziata, 98169 Messina, Italy; giada.giambrone@studenti.unime.it (G.G.); gcatone@unime.it (G.C.); marinog@unime.it (G.M.); enrico.gugliandolo@unime.it (E.G.); renato.miloro@studenti.unime.it (R.M.); 2Department of Chemical, Biological, Pharmaceutical and Environmental Sciences, University of Messina, Viale Ferdinando Stagno D’Alcontres, 31, 98166 Messina, Italy

**Keywords:** standing laparoscopic ovariectomy, equids, loco-regional pain management

## Abstract

**Simple Summary:**

Laparoscopy is a minimally invasive surgical technique used to explore and treat conditions within the abdomen. In equids, this procedure is generally performed standing, with animals sedated and restrained in stocks. Laparoscopic ovariectomy in equids has gained popularity as it avoids the risk of general anaesthesia, greatly improves visualisation and manipulation of the ovary and its blood supply, reduces the recovery period, and provides a superior aesthetic result. Indications for ovariectomy include neutering, fertility problems, ovarian tumours, hematomas or cysts, disorders of sexual development, and the elimination of undesirable behaviour related to oestrus. During the procedure, pain management is achieved with a combination of systemic or loco-regional anaesthesia. The objective of this systematic review was to evaluate studies in the equine veterinary literature, published between 2003 and 2023, in which loco-regional anaesthesia was used during standing laparoscopic ovariectomy in equids to assess the different drugs, techniques, and outcomes.

**Abstract:**

Laparoscopic ovariectomy is generally performed with equids in the standing position, with the animals heavily sedated and restrained in stocks. This procedure may be quite painful, and it is essential first to manage intraoperative pain to complete the surgery, respecting the animal’s welfare and, at the same time, ensuring the safety of the operators. Laparoscopy requires multiple small incisions to introduce the instruments, with one to two incisions enlarged sufficiently to remove the ovary. The surgical procedure must be associated with effective pain control, usually obtained with loco-regional anaesthesia, mesovarian injection, mesovaric or ovarian topical anaesthesia, and epidural anaesthesia. This systematic review aims to discuss articles published from 2003 to 2023 on treating loco-regional anaesthesia in standing laparoscopic ovariectomy in association with an evaluation of pain. The literature review was undertaken according to the Preferred Reporting Items for Systematic Reviews and Meta-Analysis (PRISMA) guidelines on three databases (NCBI-PubMed, Web of Science, and SciVerse Scopus). Despite the collected papers numbering 36, we identified only five eligible papers, demonstrating that few studies are performed in order to evaluate the quality of analgesia with loco-regional anaesthesia in standing laparoscopic ovariectomy in equids. The authors of this systematic review agree that the association of injectable and epidural anaesthesia is the best solution to manage intraoperative pain in standing laparoscopic ovariectomy in equids.

## 1. Introduction

Laparoscopy is a minimally invasive endoscopic surgical technique performed in the abdomen or pelvis. Equine laparoscopy was initially reported in 1970, but it was not until 1990 that it became commonly used for surgical intervention. Today, it is considered the standard of care in many surgical techniques, such as ovariectomy [1]. 

Indications for unilateral ovariectomy in female equids are ovarian tumours and ovarian hematomas or cysts, while bilateral ovariectomy is performed for neutering, fertility problems, or to eliminate abnormal behaviours that are hormone-related [2]. 

Although various surgical approaches are used for unilateral or bilateral ovariectomy in equids [3,4,5,6,7,8], standing laparoscopic ovariectomy (LO) under sedation is the preferred method in these species. It is a safe and reliable technique that improves intraoperative observation and manipulation of the viscera, and is associated with less morbidity and mortality than the approaches performed in dorsal recumbency under general anaesthesia [9,10]. 

Briefly, the standard approach for standing laparoscopy requires a small skin incision at the level of the paralumbar fossa. A cannula with a blunt trocar is advanced through the skin incision to allow for carbon dioxide insufflation within the abdominal cavity. Once the abdomen is distended, one or more skin incisions can be performed to advance other trocars within the abdominal cavity. The laparoscope is placed through the first dorsal cannula and connected to the light source and video camera. Once the ovariectomy is achieved, the instrument ports are enlarged sufficiently to remove the ovary. At the end of the procedure, the abdomen is deflated, the cannula isremoved, and the incisions are closed routinely [1].

The standing laparoscopy technique avoids the use of general anaesthesia. To limit potential complications during the procedure, the horse should stand still as much as possible while not being overly sedated, which could result in a recumbency risk. To perform the surgery in a standing position, a combination of sedation and local anaesthetic is required [1]. 

Specifically, during standing ovariectomy, an important goal is to combine sedative and analgesic drugs along with analgesia of the ovary, which may be performed through various techniques and anaesthetic agents [11,12].

A variety of options are available to achieve ideal sedation in horses. The most common option includes a bolus of alpha-2 agonists and opioids, followed by a variable-rate infusion of α-2 agonists [1,13]. However, standing LO can be painful, particularly when the ovary is grasped with forceps for manipulation and when the ovarian pedicle is ligated or severed through a cutting and vessel-sealing device [14].

Analgesia of the ovary could be obtained with loco-regional anaesthesia, such as an epidural, direct injection of the ovarian pedicle with analgesic drugs, or a combination of both [1,2,10].

Considering the importance of good analgesia for pain management and to avoid any possible complications during the procedure, this systematic review aims to evaluate studies published between 2003 and 2023 in the equine veterinary literature that assess the quality of analgesia with loco-regional anaesthesia during LO.

## 2. Materials and Methods

This systematic review was conducted following the “Preferred Reporting Items for Systematic Reviews and Meta-Analysis” (PRISMA) guidelines for systemic reviews [15]. A comprehensive literature search was performed, from 1 January 2003 to 31 December 2023, for manuscripts relating to equine LO. NCBI-PubMed, Web of Science, and SciVerse Scopus were used exhaustively as databases. For all the databases, the search strings were meticulously crafted and searched using the following terms: [(LO OR gonadectomy) AND (horse OR mare OR equid OR donkey OR mule) AND (loco-regional anaesthesia OR pain OR analgesia OR analgesia assessment OR analgesic drugs)]. 

Only English-language peer-reviewed papers published between 2003 and 2023 were considered. Specifically, the articles included discuss standing LO in equids, in which the quality of analgesia during and after LO was assessed. Therefore, articles were excluded if they were not written in the English language, if the LO was performed in general anaesthesia, and if the quality of the analgesia was not assessed. Single case report study design and review articles, simulations or cadaveric studies, and articles using standing ovariectomy via colpotomy or laparotomy were also excluded. Editorials, proceedings, and meeting abstracts were not included. 

Additionally, duplicated results were removed, and the authors read the titles, abstracts, and/or full text of the publications to determine their study eligibility. Eligibility was assessed following the objectives modified from “PICOs” [16]: Population: equids receiving ovariectomy; Intervention: standing LO; Outcome: degree of analgesia obtained with loco-regional anaesthesia. 

Full-text papers were accessed from university libraries, library journal subscriptions, and open-access sources. Papers that could not be retrieved were removed.

## 3. Results

### 3.1. Selection of Sources of Evidence

The total number of collected papers was 51. Specifically, 17 papers were retrieved from NCBI-PubMed, 28 from Web of Science, and 6 from SciVerse Scopus. Duplicates (*n* = 15) and those papers that were not considered eligible according to PICOs were removed. Finally, only five papers were included as eligible. A flowchart modified from the “Preferred Reporting Items for Systematic Reviews and Meta-Analyses” (PRISMA) guidelines [17] is provided to outline the process by which the search results were narrowed to the five articles included in this systematic review (Figure 1).

### 3.2. Synthesis of Results

All the included studies are original research (5/5). Among these, four studies focused on horse mares and one on mule mares. The surgical procedure was similar in all the articles. Considering the type of loco-regional anaesthesia, two articles evaluated only injectable anaesthesia, one evaluated epidural alone, and the remaining two evaluated the combination of both. The pain evaluation was performed intraoperatively in four out of five studies and postoperatively in one study. The pain scales used were the visual analogue scale (VAS) (3/5), composite pain scale (CPS), and horse grimace scale (HGS) (1/5), a system modified from Sampaio et al. [18] and Schauvliege et al. [19] (1/5). These results are summarised in Table 1. 

## 4. Discussion

This systematic review aimed to investigate studies between 2003 and 2023 evaluating the quality of analgesia with loco-regional anaesthesia in standing LO. Three databases allowed for a comprehensive search of all the potentially relevant literature, identifying only five eligible manuscripts for this systematic review.

An important limitation of the study could be the lack of evaluation of the risk of bias and quality of research; however, these went beyond our aims.

The most studied species was equines, with only one paper focused on mules and no studies found on donkeys. Two studies used an association of injectable and epidural anaesthesia. The sites for injectable analgesia included in the different studies were the mesovarium and/or the ovary or the ovarian pedicle. The drugs used were lidocaine, bupivacaine, and mepivacaine hydrochloride. Lidocaine was always used with epidural anaesthesia.

Specifically, Farstvedt et al. [14] compared intraovarian versus mesovarium infiltration of lidocaine in horses. Briefly, 15 adult mares were subjected to standing LO after sedation and epidural anaesthesia with detomidine hydrochloride (40 µg/Kg). For each mare, 2% lidocaine (10 mL) was injected into the ovary, and saline (0.9% NaCl) solution (10 mL) was injected into the mesovarium on one side, with saline solution injected into the ovary and 2% lidocaine injected into the mesovarium on the other side. The surgical procedure was performed 15 min after the lidocaine infiltration. Intraoperative pain responses were recorded during the following surgical steps: after grasping of the ovary with 2 × 3 claw forceps, during sharp dissection of the mesosalpinx, during tightening of the first loop ligature, during tightening of the second loop ligature, and transection of the ovarian pedicle. The VAS was used to describe the severity of the pain. A significant association between the presence of pain and the injection site was detected, with fewer horses reported to show signs of pain following mesovarian injection. The VAS score was significantly lower following mesovarian injection during tightening of the first and second loop ligatures than following the intraovarian injection of lidocaine. Therefore, the results of this study showed that the mesovarian injection of lidocaine was associated with significantly lower pain responses compared with intraovarian injection. The authors attribute these results to the site of deposition of the local anaesthetic. In fact, with a mesovarian injection, the anaesthetic agent is deposited directly into the tissue containing the ovarian nerve plexus.

In contrast, intraovarian injection relies on the uptake of the anaesthetic into the vasculature to indirectly desensitise the ovarian pedicle. Additionally, in an ovarian injection, the surgeon may inject the drug into a follicle or the parenchyma. The latter is highly vascular, while the basal membrane and granulosa layers of the follicle are avascular, limiting the uptake of lidocaine [24]. Regardless of the loco-regional anaesthesia, epidural administration of detomidine at doses <40 µg/kg was not reported to induce caudal analgesia [25]. Hence, it is unlikely to involve the ovarian nerve plexus. Therefore, in this study, analgesic effects should be attributed only to injectable anaesthesia, without the influence of epidural anaesthesia.

The injection sites in another study using 2% lidocaine (10–15 mL) as an injectable anaesthetic were both the ovary and mesovarium [20].

Two other studies used only injectable analgesia. Pezzanite et al. [23] compared the local effect of liposomal bupivacaine versus bupivacaine hydrochloride. One of the most common drawbacks of most local anaesthetics is their duration of action, which does not exceed 8–12 h [26]. Even if long-term delivery systems have been investigated, these systems are associated with several potential complications. For example, wound infiltration catheters may be associated with unintended placement, local oedema formation, infection, and a lack of patient compliance in the maintenance of the apparatus [27,28]. Liposomal-encapsulated bupivacaine avoids these complications, extending the duration of the analgesic effect. In this study, 15 mares were divided into three groups, treated, respectively, with 70 mL bupivacaine hydrochloride (0.75% BHCl) alone, and two different concentrations of liposomal bupivacaine (LB). Specifically, one group was treated with 0.75% BHCl (30 mL) followed by LB 20 mL, and the volume was expanded to 80 mL with saline solution (1:4 volume expansion). The other group was treated with 0.75% BHCl (30 mL) and LB 40 mL, and the volume was expanded to 80 mL with saline solution (1:2 volume expansion). The injections were performed in the mesovarium and ovary. Pain was evaluated in the postoperative period with CPS [29,30] and HGS [31,32]. The results showed that the pain scores were improved in liposomal bupivacaine-treated horses and that this effect was dose-dependent. These results could be attributed to the slow local release of anaesthetic from liposomal-encapsulated bupivacaine throughout 72 h, providing antinociception for an extended duration [33,34,35,36]. However, a liposomal-encapsulated drug should not be mixed with other local anaesthetics (except BHCl at a 1:1 mg: mg dose), as this may result in the rapid release of the bupivacaine from the liposomes [37]. 

Although commonly used, the direct injection of drugs in the ovarian pedicle could be associated with some disadvantages, such as haemorrhage within the pedicle from injury to vascular structures or the inadvertent penetration of viscera if the horse reacts during the insertion. Moreover, ovarian analgesia cannot be achieved in the event of improperly positioned needles, or haemorrhage [2]. To avoid these drawbacks, Koch et al. compared pain-related responses in mares receiving topical or injected anaesthesia of the ovarian pedicle with 12 mL of mepivacaine hydrochloride (0.4–0.5 mg/kg) in two groups of fifteen mares. The intraoperative pain was evaluated with the VAS at the time of the ovary grasping with traumatic forceps or during the activation of the vessel sealer and divider on the mesosalpinx or mesovarium. The VAS was associated with the measurements of serum cortisol concentration with jugular venipuncture performed immediately before the start of surgery, at the time of the portal anaesthesia, at the first grasp of the ovary, at the immediate conclusion of the surgery, and hourly for 5 h postoperatively. The VAS and serum cortisol concentrations did not show significant differences between the two groups of animals. Therefore, the results showed that mepivacaine direct irrigation on the ovarian surface provided intraoperative analgesia comparable to the injection of mepivacaine into the ovarian pedicle [21]. This confirms what has already been shown in dogs, where lidocaine splash is sufficient to provide analgesia during ovariohysterectomy procedures [38]. 

Among the studies evaluating epidural anaesthesia, only one did not combine this technique with local anaesthesia. Vullo et al. [22] tried to determine the analgesic efficacy and safety of a caudal epidural association of 0.2 mg/kg of lidocaine (2%) and 0.17 mg/kg of xylazine (2%), diluted in a 10 mL saline solution 0.9%, in eight standing mules undergoing elective bilateral LO. Analgesia, depth of sedation, and ataxia were scored during surgery using a scoring system modified by Sampaio et al. [18] and Schauvliege et al. [19]. Although epidural anaesthesia allowed for successful ovarian manipulation in six mules without any signs of discomfort during the entire procedure, in two mules, this technique was not effective, leading to a supplemental IV dose of xylazine in one case, or the additional infiltration of the ovarian pedicle with lidocaine 2%. The authors justified these differences by relating them to the body length of the mules. Even if there is limited information regarding the exact sensory tract of the ovary in horses, the ovarian plexus is reported to enter the caudal mesenteric ganglion, located ventral to the lumbar spinal vertebrae 3 (L3) [39]. The two mules had a greater body length than the other mules included in the study. It is possible that in those two patients, the epidural drugs did not reach the ovary sensory tract. As already reported in small animals [40,41,42], it is also possible that in equids the dose of agents administered via the epidural route should consider the extension of the vertebral column according to the desired effect [22]. However, given the frequency trend of mules requiring supplemental intraoperative xylazine or lidocaine infiltration, no significant differences were found. Therefore, a caudal epidural block allowed surgery to be easily completed in six of eight mule mares. Still, the authors suggest that additional studies are needed to establish the epidural doses of xylazine, resulting in reliable abdominal pain control in mules for standing ovariectomy [22].

Morphine sulfate is also used for epidural anaesthesia in association with injectable analgesia. Although Van Hoogmoed et al. [43] did not use a pain scale in their study, they demonstrated a reduced need for premedication drugs and shorter surgery times. However, the latter study may have been affected by a partial variation in the laparoscopic technique.

Epidural administration of drugs can be associated with some adverse effects. Drugs such as alpha-2 agonists or opioids can cause sedation if absorbed systemically. Severe ataxia and recumbency in horses can also occasionally be caused by standard doses of epidural anaesthetics [44]. The spread of local anaesthetic too far cranially can lead to paralysis of the lumbosacral nerves in pregnant mares or obese horses because of the narrowed epidural space [45]. 

Additionally, epidural analgesia may be affected by an improper injection technique, anatomic abnormalities, and fibrous adhesions from previous epidural injections, leading, for example, to a unilateral blockade [46,47]. 

Neurotoxicity caused by epidural solutions seems rare, since the local anaesthetics are only mildly acidic [46,48]. However, large volumes injected in the epidural space may cause pain in the spinal canal of horses due to compression of the sacral and lumbar nerves [49]. 

Considering the aforementioned risks of the epidural administration of drugs, Virgin et al. compared the use of continuous IV infusion or epidural detomidine hydrochloride in 12 mares. Detomidine was either administered via IV and titrated to effect from a 1 L bag of polyionic fluid at a concentration of 20 mg/L or injected into the epidural site at dose of 40 mg/kg diluted with saline solution (total volume 0.027 mL/kg). The VAS was used to grade pain during the initial grasp of the left ovary, initial grasp of the right ovary, post-injection grasp of the left ovary, ligation of the left ovary, transection of the left ovary, post-injection grasp of the right ovary, ligation of the right ovary, and transection of the right ovary. The VAS was associated with the level of serum cortisol from blood samples collected 10 min preoperatively, after the removal of the second ovary, and 10 min postoperatively. The results showed that continuous IV detomidine infusion could be a good adjustable alternative to caudal epidural detomidine, with similar analgesic effects. The only difference in pain was detectable with the VAS during the initial grasp of the left ovary, with a greater score in the group with IV detomidine infusion. The authors related this result to the shorter and more profound sedation obtained with caudal epidural anaesthesia [20]. 

The most widely used pain scale in the studies included in this review is the VAS, used alone in one study [14] and in conjunction with perioperative cortisol measurement in two studies [20,21]. The VAS originates in human medicine and is an intuitive and rapid scale. It is based on a horizontal 10 cm line, representing pain intensity that increases from none at the beginning of the line to the worst pain at the end of the line. The pain score is then read off as millimetres from zero to the end of the scale [50]. In equine medicine, the VAS score can be influenced by the amount of time taken to observe the horse, and inter-observer agreement tends to be suboptimal, particularly toward the middle and lower end of the pain scale [51]. The association of the VAS with the measurement of endogenous stress-mediating hormones, such as cortisol, could increase the sensitivity of the pain scale by making it less operator-dependent [50]. However, the relationship between stress and endocrine measures may reflect stress responses not induced by painful stimuli [52,53,54]. Therefore, these parameters are not recommended as indicators of pain in equids [50]. The composite pain scale (CPS) includes multiple variables, such as behavioural and physiological variables, or both, scored individually using well-defined classes using simple descriptive scales, and then combined to provide an overall CPS score [50]. Despite the lack of rigour in the methodology for CPS construction, it appears that the CPS is superior to the VAS, ensuring high inter-observer reliability of pain scores [55]. However, the CPS methods require experienced and/or trained observers, and more time is needed for repeated evaluation [56]. On the other hand, the horse grimace scale (HGS) is an equine pain scale based on facial expressions [57]. The association of the CPS and HGS could be a solution to evaluate postoperative pain more accurately than other systems alone. The system modified by Sampaio et al. [18] and Schauvliege et al. [19] evaluates horse behaviour and response to surgical stimuli intraoperatively [22]. Specifically, Sampaio et al. evaluated the association of midazolam and lidocaine to induce caudal analgesia and cervical dilation to facilitate endometrial biopsy procedures. The scoring system involves the evaluation of analgesia and response to noxious stimuli, the behaviour of the horse, and motor function, evaluating motor blockade and the effect on the reproductive tract, such as cervical relaxation [18]. Schauvliege et al. proposed a scoring system to assess position/ataxia, sedation depth, and surgical condition in standing horses [19]. The modified system used by Vullo et al. considers analgesia in relation to surgical stimuli, sedation depth, and ataxia [22].

From the above, it is clear that there is still no agreement on the pain scales to be used in equine medicine. All scales should first be validated for the type of procedure to which they will be applied. However, it is understood that the CPS and HGS could be useful tools to quantify pain as objectively as possible [22]. 

## 5. Conclusions

Considering the increasing focus on pain management and control, it is understood that good analgesia is indispensable during standing laparoscopic surgeries, such as ovariectomy. The aim of this review was to identify papers in which the quality of analgesia during and after equids in standing LO was assessed.

We have demonstrated that, in the peer-reviewed literature analysed over a period of 20 years, only a few studies adequately evaluate loco-regional anaesthesia in standing LO in relation to pain scales. 

Specifically, although some articles use only one of either injectable or epidural anaesthesia, the best results seem to be given by an association of these two techniques in standing LO. In addition, mesovarian injection anaesthesia appears to be superior to ovarian anaesthesia, but topical mesovaric anaesthesia could also be a possible alternative. Liposomal-encapsulated drugs could be helpful to increase analgesia duration. Moreover, in cases where epidural anaesthesia cannot be achieved, continuous IV administration of detomidine would lead to equivalent results in pain management.

Since the sample size and limitations of the data presented preclude definitive conclusions, a direction for future research is strongly suggested. For these reasons, this review emphasises the need for further useful investigations in which researchers standardise equine pain scales, providing an accurate quantification of perioperative pain in equids submitted in standing LO.

## Figures and Tables

**Figure 1 animals-14-02306-f001:**
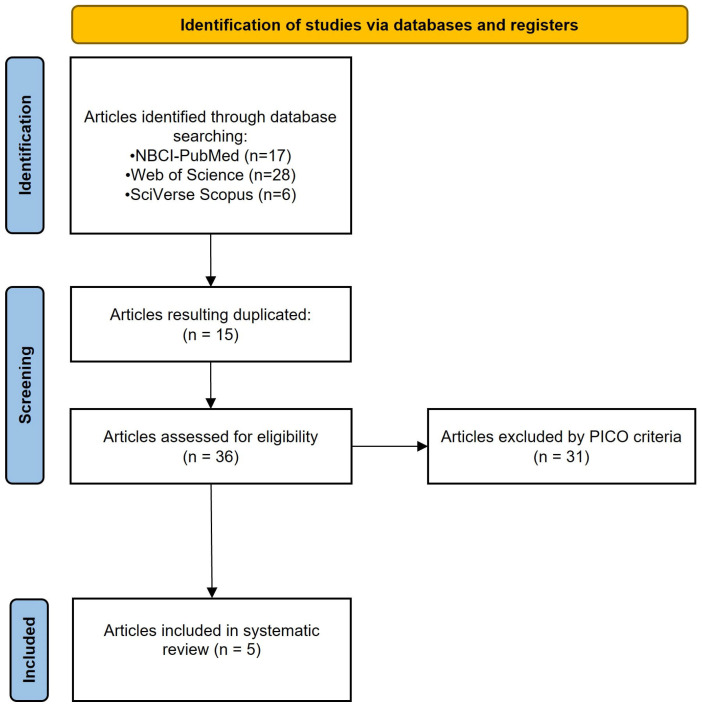
Flowchart modified from the “Preferred Reporting Items for Systematic Reviews and Meta-Analysis (PRISMA) guidelines”, showing the process of inclusion of the five articles in this systematic review.

**Table 1 animals-14-02306-t001:** Summary of the data collected from the manuscripts included in this systematic review.

Reference	Year	Species	Loco-regional Anaesthesia	Anaesthetic Drugs	Evaluation Pain Period	Pain Scale
Farstvedt et al. [14]	2005	Horse	Injective + Epidural	Lidocaine Detomidine hydrochloride	Intraoperative	VAS *
Virgin et al. [20]	2010	Horse	Injective + Epidural	Lidocaine Detomidine hydrochloride	Intraoperative	VAS * + Serum cortisol
Koch et al. [21]	2020	Horse	Injective	Mepivacaine hydrochloride	Intraoperative	VAS * + Serum cortisol
Vullo et al. [22]	2021	Mule	Epidural	Lidocaine-Xylazine	Intraoperative	System modified from Sampaio et al. [18]. and Schauvliege et al. [19]
Pezzanite et al. [23]	2022	Horse	Injective	Liposomal bupivacaine (20 mL) Vs Liposomal bupivacaine (40 mL) Vs Bupivacaine	Postoperative	CPS ** + HGS ***

* VAS = visual analogue scale; ** CPS = composite pain scale; *** HGS = horse grimace scale.

## Data Availability

Dataset available on request from the authors.

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
