# Peer review of "Loco-Regional Anaesthesia during Standing Laparoscopic Ovariectomy in Equids: A Systematic Review (2003–2023) of the Literature"

_animals, 2024, doi:10.3390/ani14162306_

Round 1

Reviewer 1 Report

Comments and Suggestions for Authors

Revision

The article “Loco-regional anaesthesia during standing laparoscopic ovariectomy in equids: a systematic review (2003-2023) of the literature.” by Giada Giambrone, Giuseppe Catone, Gabriele Marino, Enrico Gugliandolo, Renato Miloro and Cecilia Vullo reviewed studies published between
2003 and 2023 in the equine veterinary literature that evaluated the quality of analgesia with local anaesthesia during standing laparoscopic ovariectomy.

The authors used the PRISMA system to select articles from the literature and analyzed and commented on those considered eligible. The article is very interesting and well structured, methods and the results are well explained and presented. Despite this, the article needs a major revision by the authors. A general reading of the article reveals sentences that could be improved to make them more comprehensible to the reader, many of which show problems that may be due to translation from the authors' native language into English.
The way in which articles are cited in the text is often unclear, with the indiscriminate use of dashes and commas between citation numbers in the text. Revise the form and the corresponding cited bibliography. The abstract needs to be revised by adding more information on the results obtained
the conclusions of the study by decreasing the introductory part. The conclusions are very long and should make clearer the objective the authors have chosen for this article.

Below you find the specific revisions to be made in the text.

Lines 38: Laparoscopic surgeries are not only performed in the abdominal cavity. Rephrase

Line 46: The abbreviation for laparoscopic ovariectomy should only be indicated the first time it appears in the text. Include it in this line and do not indicate it every time you write these words in full.

Line 50: The two citations shown are consecutive in your enumeration, they should be separated by a comma. The dash indicates a number interval. Check this and all other references in the article applying this method.

Line 52: Change “in the skin” with “of the skin”.

Lines 58-60: This sentence is long and difficult to understand. Please rephrase it.

Line 61: Change “standing up” with “in standing position”.

Line 71: Check reference.

Line 80: Why do you write 'until 8 March 2024' if your inclusion criterion is the period 2003-2023?

Define the criteria more clearly with precise dates.

Line 83: The sentence is difficult for the reader, please rephrase it.
Line 90: Delete “the” before “English”.

Line 90: Line 87 states that only articles written in English were considered. Why is writing in English also mentioned as an exclusion criterion?

Line 91: Delete “the” before “quality”.

Line 92: Change “a simulation or cadaveric study” into “simulations or cadaveric studies”.

Line 101: How many documents were excluded because they were inaccessible? and why were they inaccessible?

Line 110: Check the number of included papers.
Fig.1: -first box of “Screening”: write “Articles resulting duplicated” and delete “published in English”
-last box of “Screening”: Change in “Articles excluded by PICO criteria”
Lines 113, 114: Change “demonstrating the process by which the search results were 113 narrowed to the 5 articles included in this systematic review.” into “showing the process of inclusion of the 5 articles in this systematic review”.

Line 119: Replace "effectuated" with "performed".
Table 1: Pezzanite et al. as explained in the discussion, compared the use of bupivacaine with two groups in which two different concentrations of liposomal bupivacaine were used. This is not stated in the table and is not understandable to the reader by consulting it. In the table (or in the description), the acronyms used should be specified to enable the reader to understand them without reading the entire article.

Lines 130, 131: Rephrase it by describing what your study’s limitations actually are.

Line 133: Replace "nothing" with " no study was found".

Line 133: The table shows that the studies using the association are 2/5, it is not definable the majority.

Line 135: Replace “ovarium” with “ovay”.

Line 141: For completeness, please also include the dosage of all drugs and if present the dilutions used in the studies you are discussing.

Line 141: The sentence is very long and unclear. Please rephrase it.

Line 148: Replace “site injection” with “injection site”.

Line 151: Delete “. lidocaine”.

Line 159: Replace “basement” with “basal”.

Line 169: replace one “main” with “principal”.

Line 171: Complete the sentence by describing the side effects discussed in your research.

Line 174: Please describe the two groups and the different dosages used.

Line 175: Why do you write "Respectively"?

Line 188: Check if the citation number 36 is correct.

Line 189: Clarify the meaning of “Topical”.

Line 191: Replace “grasping of the ovary” with “ovary grasping”.

Lines 192-196: Rephrase this sentence.

Line 216: Check the bibliography cited.

Line 216: Delete “the” before “epidural”.

Lines 244,245: Replace “amputation” with “transection”.

Lines 259-261: Rephrase this sentence.

Line 291: Replace “standing laparoscopic” with “standing laparoscopic surgeries, such as ovariectomy”.

Comments on the Quality of English Language

fine

Author Response

Dear Reviewer,

Thank you very much for taking your time to review our manuscript. We hope we have clarified what is required by following the changes suggested. In addition, the manuscript reviewed by an experienced English-speaking colleague. Thank you for your contributions that allowed us to improve the paper.

Best regards

The author’s

Revision 1

The article “Loco-regional anaesthesia during standing laparoscopic ovariectomy in equids: a systematic review (2003-2023) of the literature.” by Giada Giambrone, Giuseppe Catone, Gabriele Marino, Enrico Gugliandolo, Renato Miloro and Cecilia Vullo reviewed studies published between
2003 and 2023 in the equine veterinary literature that evaluated the quality of analgesia with local anaesthesia during standing laparoscopic ovariectomy.

The authors used the PRISMA system to select articles from the literature and analyzed and commented on those considered eligible. The article is very interesting and well structured, methods and the results are well explained and presented. Despite this, the article needs a major revision by the authors. A general reading of the article reveals sentences that could be improved to make them more comprehensible to the reader, many of which show problems that may be due to translation from the authors' native language into English.

The way in which articles are cited in the text is often unclear, with the indiscriminate use of dashes and commas between citation numbers in the text. Revise the form and the corresponding cited bibliography.

The abstract needs to be revised by adding more information on the results obtained

Ok, revised.

the conclusions of the study by decreasing the introductory part. The conclusions are very long and should make clearer the objective the authors have chosen for this article.

We tried to make clearer the aim of the study

Below you find the specific revisions to be made in the text.

Lines 38: Laparoscopic surgeries are not only performed in the abdominal cavity. Rephrase

Thank you for the comment, we added “abdomen or pelvis”.

Line 46: The abbreviation for laparoscopic ovariectomy should only be indicated the first time it appears in the text. Include it in this line and do not indicate it every time you write these words in full.

We modified as suggested

Line 50: The two citations shown are consecutive in your enumeration, they should be separated by a comma. The dash indicates a number interval. Check this and all other references in the article applying this method.

Ok, we checked all references.

Line 52: Change “in the skin” with “of the skin”.

Ok, done.

Lines 58-60: This sentence is long and difficult to understand. Please rephrase it.

We modified the sentence as suggested.

Line 61: Change “standing up” with “in standing position”.

Ok, done.

Line 71: Check reference.

Ok, done

Line 80: Why do you write 'until 8 March 2024' if your inclusion criterion is the period 2003-2023? Define the criteria more clearly with precise dates.
We removed “until 8 March”, and we added “from 1st Jannuary 2003 to 31st December 2023”

Line 83: The sentence is difficult for the reader, please rephrase it.

Ok, done

Line 90: Delete “the” before “English”.

Ok, done.

Line 90: Line 87 states that only articles written in English were considered. Why is writing in English also mentioned as an exclusion criterion?
We used this assertion based on other review that considered only articles written in English

Line 91: Delete “the” before “quality”.
Ok, done.

Line 92: Change “a simulation or cadaveric study” into “simulations or cadaveric studies”.

Ok, done.

Line 101: How many documents were excluded because they were inaccessible? and why were they inaccessible?

Again, based on other reviews, this possibility was specified but in fact all of our articles were accessible

 Line 110: Check the number of included papers.

Ok, done

Fig.1: -first box of “Screening”: write “Articles resulting duplicated” and delete “published in English”
-last box of “Screening”: Change in “Articles excluded by PICO criteria”

Thank you for your suggestion. We modified the figure according to your recommendation.

Lines 113, 114: Change “demonstrating the process by which the search results were 113 narrowed to the 5 articles included in this systematic review.” into “showing the process of inclusion of the 5 articles in this systematic review”.
Done, thank you for the comment.

Line 119: Replace "effectuated" with "performed".

Ok, done.

Table 1: Pezzanite et al. as explained in the discussion, compared the use of bupivacaine with two groups in which two different concentrations of liposomal bupivacaine were used. This is not stated in the table and is not understandable to the reader by consulting it. In the table (or in the description), the acronyms used should be specified to enable the reader to understand them without reading the entire article.

Thank you for your comment. We added the two concentrations of liposomal bupivacaine in the table. In addition, we have specified the acronyms used below the table.

Lines 130, 131: Rephrase it by describing what your study’s limitations actually are.

Ok, done

Line 133: Replace "nothing" with " no study was found".

Ok, done

Line 133: The table shows that the studies using the association are 2/5, it is not definable the majority.

Thank you for your comment. We changed the sentence.

Line 135: Replace “ovarium” with “ovay”.

Ok, done

Line 141: For completeness, please also include the dosage of all drugs and if present the dilutions used in the studies you are discussing.

Thank you for your suggestion. We added the dosages and dilutions of drugs used in the studies included in the review

Line 141: The sentence is very long and unclear. Please rephrase it.

Ok, done

Line 148: Replace “site injection” with “injection site”.

Ok, done.

Line 151: Delete “. lidocaine”.

Ok, done.

Line 159: Replace “basement” with “basal”.

Ok, done.

Line 169: replace one “main” with “principal”.

Ok, done.

Line 171: Complete the sentence by describing the side effects discussed in your research.

Thank you for your suggestion. We added these information.

Line 174: Please describe the two groups and the different dosages used.

Thank you for your suggestion. We better described the two groups.

Line 175: Why do you write "Respectively"?

We removed it

Line 188: Check if the citation number 36 is correct.

Ok, done

Line 189: Clarify the meaning of “Topical”.

Ok, thak you for your comment. We added “direct instilled on the ovarian surface”

Line 191: Replace “grasping of the ovary” with “ovary grasping”.

Ok, done

Lines 192-196: Rephrase this sentence.

Ok, done

Line 216: Check the bibliography cited.

Thank you for your comment. We made a mistake reporting the bibliography. We have now included the correct reference articles.

Line 216: Delete “the” before “epidural”.

Ok, done.

Lines 244,245: Replace “amputation” with “transection”.

Ok, done.

Lines 259-261: Rephrase this sentence.

Ok, done

Line 291: Replace “standing laparoscopic” with “standing laparoscopic surgeries, such as ovariectomy”.

Ok, done.

Reviewer 2 Report

Comments and Suggestions for Authors

Equine laparoscopy is minimally invasive, avoids the risk of general anaesthesia and shortens the convalescence period. Today it is considered the standard method in many surgical procedures such as ovariectomy.   This systemic review covers the articles published from 2003 to 2023 treating locoregional anaesthesia in standing laparoscopic ovariectomy in association with an evaluation of pain. The total number of collected papers was 51. Application of PRISMA guidelines based on three databases reduced total number to only  5 eligible papers. So the entire review is based only on these 5 reliable publications. In the references there are 58 positions. It is not clear whether among them there are also all excluded articles.  However the number of collected data is limited,  there are some important conclusions provided by the authors: good analgesia is of a great importance during standing laparoscopic, such as ovariectomy. The best results can be obtained where both injective and epidural anasthesia are implemented. Moreover, mesovarian injection anasthesia appears to be superior to ovarian anaesthesia. What is disturbing  in this review is the high risk of bias and poor quality of research based on only 5 papers.  For example , in 4 papers there is intraoperative pain evaluation and  in only one postoperative evaluation. Only in one paper authors use composite pain scale (CPS) together with horse grimace scale (HGS). In remainig four there is visual analogue scale (VAS) . These methods are so different that cannot be combined together.  In summary, review is interesting but not very convincing. It can be published but after a major revision, The reviewer recommends to collect more data which would reinforce the conclusions.

Author Response

Dear Reviewer,

Thank you very much for your constructive and appreciated observations.

Although we completely agree with your comments, we would like to highlight that, since equine standing laparoscopic ovariectomy (LO) has become surgical procedure widely diffused, many anesthetic protocols have been investigated to perform this technique (as described by many articles founded in the literature). Despite this, the reports that objectively focuses on the antalgic effects of different anesthetic techniques are few in equids, compared to those that has been widely described in small animals for the same laparoscopic procedure. Our review wants to emphasize the importance of the objective assessment of pain by means of different equine acute pain scales in order to evaluate the quality of the singular analgesic techniques using during LO.

We do not believe that the quality of this research is poor, rather we believe that in 20 years there has been little focus by researchers on the evaluation of analgesia despite the many sedative protocols proposed for standing LO in equids. This is indeed evident from the only 5 eligible papers.

We agree that only in one paper did authors use the composite pain scale (CPS) together with the horse grimace scale (HGS) and that in the remaining four only the visual analogue scale (VAS) is used. However, during the drafting of a paper, information is taken from articles already published, despite the research method may be questionable.

Could you agree to add these considerations to the conclusions? Which, in any case, have been modified as suggested.

Finally, regarding the high risk of bias, in the discussion we declared that a limitation of the study could be the lack of evaluation of the risk of bias and quality of research, despite these going beyond the aims of this study.  

We hope to have clarified your doubts about this paper.

Best regards

The author’s.

Round 2

Reviewer 1 Report

Comments and Suggestions for Authors

Dear authors, 

Thank you for your work and for editing the paper according to my suggestions. 

Last point to be revised is the formatting of the bibliography numbering.

Best regards  

Author Response

Dear Reviewer,

Thank you for taking the time to work on our manuscript.

We revised the formatting of the bibliography numbering as suggested.

Best regards

The authors.
